# Changes in Active Behaviours, Physical Activity, Sedentary Time, and Physical Fitness in Chilean Parents during the COVID-19 Pandemic: A Retrospective Study

**DOI:** 10.3390/ijerph19031846

**Published:** 2022-02-06

**Authors:** Sam Hernández-Jaña, Danica Escobar-Gómez, Carlos Cristi-Montero, José Castro-Piñero, Fernando Rodríguez-Rodríguez

**Affiliations:** 1IRyS Group, Physical Education School, Pontificia Universidad Católica de Valparaíso, Valparaíso 2581967, Chile; sam.hernandez@pucv.cl (S.H.-J.); carlos.cristi.montero@gmail.com (C.C.-M.); 2GALENO Research Group, Department of Physical Education, Faculty of Education Sciences, University of Cádiz, Puerto Real, 11003 Cádiz, Spain; jose.castro@uca.es; 3Biomedical Research and Innovation Institute of Cádiz (INiBICA) Research Unit, 11009 Cádiz, Spain

**Keywords:** healthy behaviour, family, SARS-CoV-2, quarantine, isolation, lockdown

## Abstract

Strategies to reduce the spread of coronavirus disease 2019 (COVID-19) have caused different behavioural modifications in all populations. Therefore, this study aimed to determine changes in active commuting, moderate-to-vigorous physical activity (MVPA), physical fitness, and sedentary time during the COVID-19 pandemic in Chilean parents. Eighty-six fathers (41.30 ± 6.82 years) and 294 mothers (40.68 ± 6.92 years) of children from different schools from Valparaíso, Chile, participated. Inclusion criteria were adults with schoolchildren who were resident in Chile during the research period. Convenience sampling was used as a non-probabilistic sampling technique. Respondents completed a self-reported online survey about active commuting, MVPA, self-perceived physical fitness, and sedentary time July–September 2020 during the first pandemic period. Comparisons between before and during the pandemic were performed using t-tests and covariance analysis (ANCOVA), establishing a significance level at *p* < 0.05. Most participants stayed at home during the pandemic, whereas active and passive commuting significantly decreased in both fathers and mothers (*p* < 0.001). MVPA and physical fitness scores reduced considerably (*p* < 0.05), while sedentary time significantly increased (*p* < 0.05), independent of the sex of parents and children’s school type. Differences by age groups and the number of children were more heterogeneous, as younger parents showed a larger decrease in MVPA (*p* < 0.05) and physical fitness score (*p* < 0.05). Additionally, parents with one child showed a larger decrease in sedentary time (*p* < 0.05) than those with two or more children. The COVID-19 pandemic significantly affected healthy behaviours. Hence, health policies should promote more strategies to mitigate the long-term health effects of the pandemic on Chilean parents.

## 1. Introduction

The severe acute respiratory coronavirus disease caused by SARS-CoV-2, commonly referred to as COVID-19, was declared to be a pandemic by the World Health Organization on 11 March 2020 [1]. Health and economic consequences have been devastating in different countries around the world [2,3], affecting all populations in different ways [4], with greater effects in some communities due to health and social inequalities [5]. Given the high risks of contagion, health policy makers established different strategies to stem the spread, with some of the restrictions remaining in place today in order to complement other public health strategies [6]. Not only has COVID-19 itself had detrimental effects on emotional health and well-being and caused cardiovascular, respiratory, and neurologic manifestations [2], but the strategies to reduce spread have also had implications on other behaviours associated with health, such as the mode of commuting, decrease in physical activity (PA), and increase in sedentary time [7,8]. It has been previously shown that low fitness increases the risk of early mortality from all causes, cardiovascular disease, and cancer [9], as well as mental health outcomes [10].

On the one hand, both PA and physical fitness have been associated with several health outcomes [11,12]. For instance, the former has been associated with lower premature mortality and is considered an effective primary and secondary preventive strategy for at least 25 chronic medical conditions [11]. In addition, physical fitness has been inversely linked to all-cause, cardiovascular, and cancer-related mortality [9]. In addition, sedentary time has been associated with multiple detrimental health outcomes such as all-cause mortality, fatal and non-fatal cardiovascular disease, type 2 diabetes, and metabolic syndrome and could be an independent risk factor from PA [13]. These factors have a number of public health implications, as they may be seriously affected by health strategies adopted to mitigate the effects of the pandemic [14]. Due to governmental health measures, freedoms were restricted, and most of the population had to stay in their homes [6], which invariably led to the deterioration of many healthy behaviours. These changes may modify some adult behaviours, and notably, directly affect those who have children, due to quarantine and isolation altering school and work routines [6,15].

Given these factors, it has been shown that parents are a key influencing factor in the behaviour of their children [16], including sedentary and active behaviour. Some reviews from before the COVID-19 pandemic have identified associations between parental support and children’s PA [17,18]. Specifically, being allowed to play anywhere in the neighbourhood, family encouragement, family social support, and family activities have been associated with out-of-school physical activity in schoolchildren [19]. On the other hand, some studies have demonstrated that active commuting to school (walking or cycling) is an opportunity to reach daily PA levels [20,21]. Additionally, a general decline in active commuting has been observed in recent decades in a number of countries [22,23,24]. This decline could be partially explained by rising distances between people’s homes and workplaces and by car availability [25]. Various family factors, such as socioeconomic status [26] and parents educational [27] and professional levels [28] may influence this. Nevertheless, the family factors that can best predict this active behaviour have not been clearly defined, especially in Latin American countries. It is crucial to know how parent behaviours may have been affected by the COVID-19 pandemic in order to design strategies to mitigate its long-term effects. To the best of our knowledge, there is no evidence regarding how healthy behaviours have changed in Chilean families, especially in parents. Therefore, this study aimed to determine whether there were changes in active behaviours and physical fitness during the COVID-19 pandemic in Chilean parents. Further, we aimed to discover the characteristics of the families that have decreased their PA the most.

## 2. Materials and Methods

### 2.1. Study Characteristics and Ethical Considerations

This retrospective study is part of the research project “Familial and scholarly environment: Effects of physical activity levels of parents on PA levels of children and adolescents”, which aims to determine the association between parents’ physical activity levels and their children’s, according to different sociodemographic variables. Prior to participation, all parents voluntarily signed a consent form that explained the aims and scope of the study. Information was uploaded to an online cloud, with only the main project researcher (F.R.-R.) having access to data. The research followed ethical principles of the Declaration of Helsinki on research on human subjects (World Medical Association 2013) [29]. Lastly, the research project was approved by the Ethics Committee of the Pontificia Universidad Católica de Valparaíso (BIOEPUCV-H 363-2020).

### 2.2. Participants

Eighty-six fathers (41 ± 6.8 years) and 294 mothers (40 ± 6.9 years) from different Chilean schools belonging to the central region of the country participated as a non-randomised sample. The recruitment process consisted of invitations to participate in the project to principals of private and public schools and through social media (Instagram, Facebook, email). Inclusion criteria were adults with school-age children who were residents in Chile during the research period. Exclusion criteria were having a cognitive impairment that prevented answering the questions, parents <18 years old, or having children aged <5 or >14. Only three participants were excluded, due to being under the required age.

During the data collection, adult participants were authorised to attend their work if it was “essential”. However, most of them worked online from home. In the case of schools, a large proportion undertook virtual classes from home and only a few private schools had children attend in person. In general, during the evaluation period, the participants did not have important mobility restrictions but were subject to restrictions in closed spaces (workplaces and schools) where only one person was allowed for every 6 m^2^.

### 2.3. Study Design and Primary Outcomes

This was a retrospective study regarding how active behaviours, PA, physical fitness, and sedentary behaviour changed during the COVID-19 pandemic in Chilean parents. To achieve the objective, a self-reported online survey was undertaken in Spanish from July to September 2020 using the SurveyMonkey platform (San Mateo, Ca, USA). Convenience sampling was used as a non-probabilistic sampling technique. Each participant completed a questionnaire about active commuting, moderate-vigorous PA (MVPA), physical fitness, and sedentary time during the first lockdown and how these factors were before the lockdown, using the same questions raised retrospectively, remembering what they were doing four months ago. The first section covered sociodemographic characteristics such as sex (male, female), age (in years), residence (urban, rural), number of children, and children’s school type (private, public). Similarly, one question evaluated their employment situation in order to assess how the pandemic changed their jobs. This question was: Has your work situation changed due to the Coronavirus pandemic? The answer options were: (1) “I commuted normally to my job”; (2) “I have always worked at home”; (3) “I stayed at home, unable to work; (4) “I can do my work almost normally from home”; (5) “I stayed at home, I was able to work at home at times”; (6) “My work was affected as there was less demand for my services”; (7) “I was fired or my source of income was suspended”; (8) “I am a homeowner, so my situation has not changed”; (9) “During the pandemic I have been on prenatal/postnatal rest”. All questions regarding active commuting, MVPA, self-reported fitness, and sedentary time included the terms “before” and “during” the pandemic to assess the change in parents’ behaviours.

Active commuting was evaluated using the fourth version of the “PACO” (Pedalea y anda al colegio) questionnaire from the Universidad de Granada [30], which evaluates how children travel to/from school. The questions about active commuting were derived from an exhaustive review of studies of the topic [31] and were shown to be reliable [30,32] and to have been validated in the Chilean population [33]. The questions were adapted to the parents’ context. These questions were: “How do you usually get to work?” and “How do you usually get home from work?”. The answer options were: walking, cycling, car, motorcycle, public bus, metro/train, or other (description was required). We did not include in the results those people who chose the option “other” method of commuting, because is a non-specific answer to stratify the mode of commute. Finally, mode of commuting was dichotomized as active or passive commuting and analysed.

MVPA and sedentary time levels were evaluated using the “Global Physical Activity Questionnaire” (GPAQ) [34]. Having performed an analysis of the validity of the IPAQ (International Physical Activity Questionnaire) and the GPAQ, the latter was chosen as the best option since it covers more aspects and has similar validity to the IPAQ [35,36]. The GPAQ was developed by the World Health Organization to collect information on PA and sedentary behaviour at work, activity on the move, and activity in leisure time through 16 closed and open questions (mixed). Physical activities with an expenditure above 4 METs are considered to be of moderate-intensity and those with an expenditure of ≥ 8 METs of vigorous-intensity.

The equation for calculating METs/min/week is = [(P2 × P3 × 8) + (P5 × P6 × 4) + (P8 × P9 × 4) + (P11 × P12 × 8) + (P14 × P15 × 4)], where P is the time, and the attached number is the question corresponding to work, transportation, or recreation PA. Question 16 (Q16) corresponds to the time spent on sedentary activities. Parents were classified as physically active when they completed ≥ 150 min/week—meeting the recommendations—and physically inactive when they did not reach 150 min/week—not meeting the recommendations [37].

Physical fitness was measured using the International Fitness Scale (IFIS), a self-reported questionnaire with five Likert-scale questions about different perceived fitness domains: overall, cardiorespiratory fitness, muscular fitness, speed-agility, and flexibility fitness [38]. This test was chosen because the restrictions of the pandemic made it impossible to evaluate fitness objectively or in a more practical context.

### 2.4. Covariates

The sex and age of parents and children’s school type were used as covariates. First, people showed changes in their behaviour during the pandemic, and these changes differed according to sex. For example, women were more likely to have permanently lost their job than men due to the pandemic [39]. Likewise, more of those who changed their mode of commuting due to COVID-19 were women [40]. Regarding physical activity, differences according to sex have previously been found, with men being more active than women [41]. Indeed, men spend more time in MVPA and sedentary time than women [42]. However, during the pandemic, men reported a larger decrease in PA and a larger increase in sedentary time compared with women [43].

Second, regarding the age of the parents, there were different strategies to reduce the spreading of the virus, such as lockdowns or telework, which modified behaviour in some groups. For instance, targeted interventions were recommended to age-specific groups to reduce the health burden of the pandemic and minimise social and economic impact, such as full lockdown [44]. On the other hand, the strategies impacted PA and sedentary time, with the adult population decreasing their time spent on vigorous PA the most, whereas youngest subjects showed a decrease in moderate PA and an increase in sedentary time [45]. Similarly, it has been found that PA levels differed according to age from childhood to adulthood and aging [45,46]; however, research has also found that the prevalence of inactivity increases with age [47].

Lastly, we used the children’s school type as a proxy variable for socioeconomic status. There were health outcome disparities according to socioeconomic status in this population during the pandemic, with subjects with lower education levels and some specific communities having strong associations with adverse effects due to COVID-19 [48]. In addition, while worldwide PA levels decreased during the pandemic, this reduction was likely influenced by socioeconomic status [49].

### 2.5. Statistical Analysis

Missing data (179 subjects had at least one missing piece of data, ranging from 3.4% to 31.0% of the total) were imputed using the random forest method using the R package (RStudio, Boston, MA, USA)“missForest” [50]. Data with sensitivity analysis are presented in the Appendix A. Continuous variables are presented as mean and standard deviation, whereas categorical variables are presented as frequency and percentage. Parametric tests were performed due to the sample size [51]. Continuous baseline characteristics were compared using Student’s t-test for independent samples, whereas baseline characteristic proportions and active commuting were compared using the chi-squared test.

Analysis of covariance (ANCOVA) was performed to compare changes in baseline values and delta values for five categories: sex of parents (men/father and women/mother), children’s school type (public and voucher schools were grouped in the “public school” category, while private schools kept the same name (“private school”)), age group (20 to 39 years and 40 to 59 years), number of children (one child or two or more children), and a combination between age group and number of children (G1: 20–39 years with one child; G2: 20–39 years with two or more children; G3: 40–59 years with one child, G4: 40–59 years with two or more children). All ANCOVA analyses were adjusted by sex and age of parents and by children’s school type, except when the variable was used to stratify categories. Moreover, the Bonferroni post hoc test was used for multiple comparisons. JAMOVI statistical software (JAMOVI Version 1.6, Computer Software, Sidney, Australia) and RStudio (RStudio, Boston, MA, USA) were used for analyses, and GraphPad Prism (GraphPad, San Diego, CA, USA) was used for graph design. Statistical significance was set at *p* < 0.05.

## 3. Results

Table 1 shows sociodemographic characteristics of the participants before the pandemic according to the sex of parents. The employment situation presented was measured only once during the pandemic. In this question, there was a significant difference (*p* < 0.001) between the response options, where the three highest prevalence’s were “*I can work almost normally from my home*” (30.5%), “*My situation has not changed*” (16.3%), and “*Normal, I travelled to my job normally*” (12.9%). On the other hand, fathers had a higher score on physical fitness compared with mothers (*p* < 0.001).

Results of baseline characteristics (before the pandemic) by sex of parents for active commuting, MVPA, physical fitness score, and sedentary time are shown in Table 2. According to the mode of commuting, three categories were created (“Stay home” when parents did not commute during the lockdown period; “Active” when parents commuted to work during the lockdown; and “Passive” when parents commuted to work during the lockdown by passive modes). For MVPA and sedentary time, parents were dichotomized for three main factors: school (private or public), age (20–39 years or 40–59 years), and number of children (1 child or ≥2 children). Regarding mode of commuting, there were significant differences between the modes of commuting “to work” by sex of the parents (*p* = 0.010) and back “to home” (*p* = 0.026). In addition, passive commuting was the most frequent mode of commuting before the pandemic (69.8%). There was only one significant difference in MVPA, with fathers with one child being more active than mothers with one child (*p* = 0.008). For sedentary time, there was one difference in the 40–59 years category, with fathers having more sedentary time than mothers (*p* = 0.027).

Figure 1 illustrates changes in the mode of commuting both “to work” and “to home” before and during the pandemic. Passive commuting was the most frequent before the pandemic for both fathers and mothers. Nevertheless, both active and passive commuting decreased during the pandemic, and those parents who stayed at home became the most prevalent (*p* < 0.001). All changes described above were statistically significant (*p* < 0.001).

Figure 2 shows the difference (Δ) in minutes in MVPA, physical fitness score, and sedentary time before and during the pandemic by sex, children’s school type, age group, and number of children. The results show that all changes were statistically significant (*p* < 0.001) compared with their baseline values (shown by asterisks). No significant differences were found by sex, type of school, MVPA, physical fitness, and sedentary time. However, the decrease in MVPA time and physical fitness was significantly greater (*p* < 0.001) in the group of younger parents (20–39 years). Likewise, parents with two or more children showed a greater decrease in the MVPA time compared with parents with only one child (*p* < 0.05). In contrast, physical fitness was lower (*p* < 0.05) and sedentary time was higher (*p* < 0.05) in parents with only one child than parents with two or more children during the pandemic.

Figure 3 displays comparisons of MVPA, physical fitness, and sedentary time by age group and number of children. All groups showed a decrease in MVPA time during the lockdown. In addition, G2 (20–39 years with 2 or more children) presented a greater loss in weekly minutes in MVPA compared with G3 (40–59 years with 1 child; *p* < 0.05) and G4 (40–59 years with 2 or more children; *p* < 0.05).

Regarding physical fitness, only G1 and G2 showed significant decreases during lockdown (*p* < 0.001), with G1 being significantly lower than G4 (*p* < 0.05). Finally, all groups showed significantly increased sedentary time during lockdown, with G3 being significantly higher than G4 (*p* < 0.05).

## 4. Discussion

This study aimed to determine changes in active commuting, MVPA, physical fitness, and sedentary time during the COVID-19 pandemic in Chilean parents. The main results were that most parents stayed at home; thus, active and passive commuting decreased significantly. MVPA and physical fitness reduced considerably, whereas sedentary time increased dramatically in all parents. While differences were significant in all participants compared with the baseline values, when parents were grouped according to age groups, those who were younger had a higher decrease in MVPA and physical fitness compared with older ones.

### 4.1. Active Commuting

Social distancing measures and lockdowns were applied in many countries [52], implying that children and adolescents and their parents should stay at home. According to our findings, parents stayed at home because they could work from there, or unfortunately, because their jobs were affected, or they were fired.

For instance, research from Philadelphia (US) found that nearly half of essential workers changed their mode of commuting due to the COVID-19 pandemic. The main reasons mentioned were safety and potential exposure to the virus [7]. Similarly, the “Canadian National Survey Data” showed that all modes of commuting declined, whereas telework increased [40], as seen in the Chilean context. In the UK, 80% of trips during the pandemic were by motorised transport, and 59% of car trips were less than 5 km [53]. This led to the UK Government recognising this as an opportunity and announcing a GBP 2 billion investment package to create a new era of active commuting [54].

Unfortunately, there is scarce evidence about how modes of commuting have changed during the COVID-19 pandemic, and this lack of knowledge seems more notable in Latin American countries. According to our results, both fathers and mothers decreased active and passive commuting.

However, an important fact is that mothers spent more time at home engaging in caregiving, educational, and play activities than fathers before the pandemic [55]. Indeed, mothers have reported that they supported children’s activities during the week because fathers used to work long hours or late into the evening [56]. Despite this, fathers and mothers uniformly increased their time at home. Undoubtedly, this lack of commuting has had a significant adverse effect on PA, considering that active commuting in many cases is the only opportunity to increase the levels of PA in order to reach the recommendations [20,21]. Likewise, those parents who had to commute to their jobs anyway kept commuting by a passive mode predominantly. This is important, because although the government implemented lockdowns, some people had special authorisation to go to work. However, we do not have information regarding this. Hence, we can only confirm that, according to our results, those parents who had to commute to their jobs mainly kept commuting in a passive way. In this regard, a previous Chilean study demonstrated an important association between parents’ active commuting to work and children’s active commuting to school, especially for mothers [57]. Therefore, this reduction in the active commuting of the parents could also affect the mode of commuting of their children. During the pandemic, local governments, companies, and schools should create strategies that promote the active commuting of parents to work and children to school as a post-lockdown compensatory measure.

### 4.2. Physical Activity

Variation in MVPA was an expected consequence in terms of behavioural modifications caused by health measures such as social distancing or lockdowns. For example, a survey of healthy adults in Spain showed that moderate and vigorous PA decreased by 2.6% and 16.8%, respectively, during the lockdown in all populations [43]. These modifications seem to be specific to certain characteristics. For instance, a Canadian survey found that 40.5% of physically inactive people decreased their level of physical activity, while for active people, only 22.4% decreased their physical activity [58].

In the current study, MVPA levels decreased, independent of sex and children’s school type. While a Spanish population survey found that men reduced vigorous physical activity during COVID-19 confinement more than women, they did not statistically compare the differences between the sexes [43], so it is not possible to determine whether the differences were significant. Similarly, the differences were more noticeable when comparing the differences by age group and the number of children per family. Younger parents and those who had one child showed a larger decrease compared with older parents or those with two or more children. In this regard, a US study found that physical activity determined by daily steps decreases with age [59]. To our knowledge, there is still no evidence on the effect of the age of parents and its effect on the decrease in physical activity during lockdown. However, this difference could be because the younger parents had a higher baseline level of physical activity (+33.1%), on which there was a greater impact compared with older parents. It has been shown that paternity is associated with difficulties in taking part in some physical activities [60,61], and these difficulties are more pronounced in parents with younger children [62]. Thus, it is probable that younger parents had younger children and that is associated with lower physical activity levels [62] because younger children are more dependent and require more parental care [63]. Similarly, data from the 2017 Behavioral Risk Factor Surveillance System showed that women’s physical activity levels were associated with the presence of children at home; however, the number of children was not associated with the effect size [64]. More studies are needed to determine whether there is a repeated pattern affecting the level of physical activity in parents in other countries and contexts.

### 4.3. Physical Fitness

It was challenging to measure physical fitness using direct or field-based tests to determine how fitness changed during the pandemic. Thus, a self-reported test was applied according to the health conditions allowed. Although people infected with COVID-19 showed impairments in various physical function indicators [65], it may be possible to deduce that physical fitness levels decreased due to less physical activity and more sedentary time [66], which was unavoidable, as health strategies limited several activities such as commuting, going to gyms, or spontaneous physical activity performed during some daily activities.

Despite the importance of an adequate physical fitness level, there is scarce evidence about its change during the pandemic, which is crucial, considering the role of physical condition in the severity of COVID-19 and other diseases [67,68]. Our research found that participants’ perception of their physical fitness levels decreased in all groups, independent of the sex of parents and children’s school type. Moreover, physical fitness differed according to the age group and number of children category, as the reduction in levels was more noticeable in younger parents and those who had one child. While both groups had a higher average value at baseline, it was expected that the reductions would affect all groups, as the survey was applied at a critical moment during quarantine in Chile. Hence, it is important to face consequences of the pandemic, assessing current physical fitness levels in different age groups to design interventions that can compensate for the impairment of physical fitness and health due to COVID-19 prevention strategies.

### 4.4. Sedentary Time

Sedentarism has already been linked to many detrimental health outcomes [65], and the COVID-19 pandemic increased time spent on several sedentary behaviours in all populations due to restrictions. A cross-sectional and retrospective online survey quantified changes in sedentary behaviours in Brazilian participants, finding that sedentary behaviour increased by 2.5 h/day, an increase of 40% [69]. Likewise, another study in young Spanish adults measured different sedentary behaviours such as sitting time and smartphone use during the COVID-19 lockdown, concluding that the participants increased their sitting time and smartphone use [8]. Our findings align with these results, as sedentary time increased, independent of sex, children’s school type, and age group. While differences according to the number of children were statistically significant, both groups differed in their baseline values, which means that lockdown affected all populations.

In addition to the above-mentioned variables, there have also been differences found between sedentarism according to some work-based activities. For instance, research on Japanese participants found that those who worked from home had more uninterrupted sedentary time than those who worked in their workplaces [70]. Similarly, a cross-sectional survey in Jordanian adults examined changes in sedentary behaviours, finding that most participants reported an increase in sedentary behaviours such as watching TV, using electronics, and using to social media [71]. Despite our research measuring sedentarism, we were limited to quantifying total sedentary time, which means we could not consider differences among specific sedentary behaviours and the health impact [72].

In this regard, researchers must create intervention strategies in parents and families to test the independence of families in maintaining physical activity and reducing sedentary time. This would allow us to acquire useful knowledge and be better prepared for complex health events that may occur in the future.

### 4.5. Strengths and Limitations

One of the main strengths of this study was the discovery of the characteristics of the parents that most influence the reduction in their physical activity and that could eventually affect the physical activity of their children. Furthermore, to the best of our knowledge, there is no evidence to date regarding active commuting, MVPA, physical fitness, and sedentary time changes during the COVID-19 pandemic in Chilean parents. Additionally, the data imputation process that was carried out improved the statistical power of the analyses.

Nonetheless, this research shows some limitations. First, the use of the physical activity self-report questionnaire was less objective than other measurement strategies such as accelerometery. Likewise, this issue affected the fitness self-report questionnaire. Second, the study design meant that cause–effect relationships cannot be determined. There could also have been a selection and recall bias due to not performing probability sampling and the study design, respectively. On the other hand, the missing data could have increased the estimation error; however, the sensitivity analysis did not show large differences. Additionally, the type of qualitative and dichotomous variables limited us to performing chi-square analysis, which may not have been the best analysis strategy. Finally, the results should be interpreted with caution because the sample selection only allows us to draw local conclusions and not valid conclusions for the rest of the country or other populations.

## 5. Conclusions

Active commuting was seriously affected, as quarantines kept people in their houses. Therefore, it is important to provide appropriate conditions for people who travel to work despite the confinement because many parents often have few chances to achieve physical activity recommendations. Likewise, physical activity levels decreased significantly, whereas sedentary behaviour increased in all populations. This change may negatively impact physical fitness, which could lead to negative impacts for public health.

Given these results and that people mostly stayed at home, future research could determine the influence of housing type on changes in healthy behaviours, which could help to promote community-contextualised initiatives to mitigate the impact of total confinement on population health.

Overall, the lockdown caused by the COVID-19 pandemic negatively affected active commuting, MVPA, and physical fitness level and increased the sedentary time of parents. These changes could, in turn, affect the child’s physical activity and sedentary behaviour. For this reason, policies and interventions should be created to compensate for the loss in levels of physical activity and physical fitness of parents and their long-term impact on the health and fitness of their families. These interventions should be focused on the most vulnerable groups, especially the youngest parents.

## Figures and Tables

**Figure 1 ijerph-19-01846-f001:**
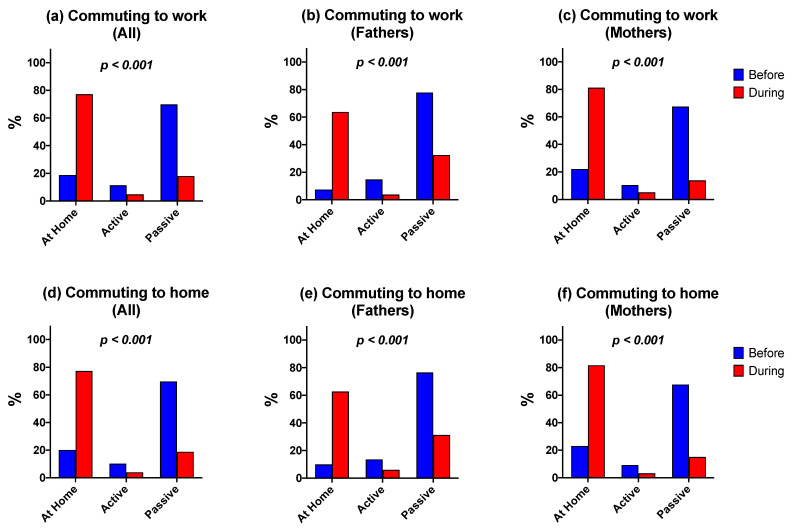
Prevalence of active commuting “to work” and “to home” before and during the COVID-19 pandemic according to mothers and fathers. (**a**) Active commuting to work before and during the pandemic in both sexes, (**b**) Active commuting to work before and during pandemic in fathers, (**c**) Active commuting to work before and during the pandemic in mothers, (**d**) Active commuting to home before and during the pandemic in both sexes, (**e**) Active commuting to home before and during pandemic in fathers, (**f**) Active commuting to home before and during the pandemic in mothers.

**Figure 2 ijerph-19-01846-f002:**
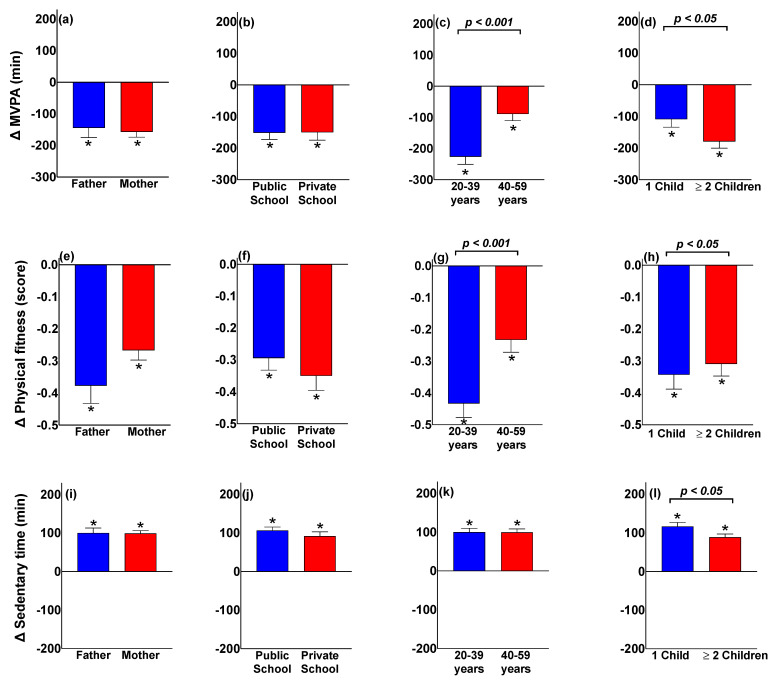
Influence of the COVID-19 pandemic on the variation in MVPA, physical fitness, and sedentary time by sex of parents, children’s school type, age group and the number of children by family. ANCOVA models were adjusted by age, sex, and children’s school type, except when the variables were used to stratify the data; MVPA: moderate-to-vigorous physical activity; * *p* < 0.05 compared with the value before the pandemic. (**a**) Variation after-during of MVPA between fathers and mothers, (**b**) Variation after-during of MVPA by children’s school type, (**c**) Variation after-during of MVPA by parents age group, (**d**) Variation after-during of MVPA by the number of children in the family, (**e**) Variation after-during of physical fitness between fathers and mothers, (**f**) Variation after-during of physical fitness by children’s school type, (**g**) Variation after-during of physical fitness by parents age group, (**h**) Variation after-during of physical fitness by the number of children in the family, (**i**) Variation after-during in sedentary time between fathers and mothers, (**j**) Variation after-during in sedentary time by children’s school type, (**k**) Variation after-during in sedentary time by parents age group, (**l**) Variation after-during in sedentary time by the number of children in the family.

**Figure 3 ijerph-19-01846-f003:**
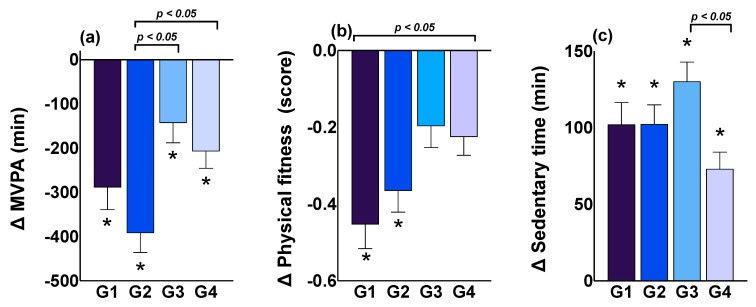
Effect of the COVID-19 pandemic on the variation in MVPA, physical fitness (score), and sedentary time by age group and number of children. ANCOVA models were adjusted by sex of parents and children’s school type; MVPA: moderate-to-vigorous physical activity; * *p* < 0.05 according to the value before the pandemic. (**a**) Variation before-during the pandemic on MVPA between four groups, (**b**) Variation before-during the pandemic on Physical fitness between four groups, (**c**) Variation before-during the pandemic on Sedentary time between four groups. G1: Group 20–39 years with 1 child; G2: Group 20–39 years with 2 or more children; G3: Group 40–59 years with 1 child; G4: Group 40–59 years with 2 or more children.

**Table 1 ijerph-19-01846-t001:** Baseline sociodemographic characteristics of participants.

Parental Sociodemographic	All (*n* = 380)	Fathers (*n* = 86)	Mothers (*n* = 294)	*p*-Value
Age (years)	40.8 ± 6.9	41.3 ± 6.8	40.7 ± 6.9	0.462
20–39 years	34.5 ± 3.9	35.2 ± 3.7	34.3 ± 3.9	0.229
40–59 years	45.7 ± 4.2	45.9 ± 4.6	45.7 ± 4.1	0.697
Children’s school type				
Public	231 (60.8%)	51 (59.3%)	180 (61.2%)	0.748
Private	149 (39.2%)	35 (40.7%)	114 (38.8%)	
Number of school-age children				
One	164 (43.2%)	35 (40.7%)	129 (43.9%)	0.600
Two or more	216 (56.8%)	51 (59.3%)	165 (56.1%)	
Employment situation				
Normal, I moved to my job normally	49 (12.9%)	21 (24.4%)	28 (9.5%)	**<0.001**
I have always worked in my home	19 (5.0%)	4 (4.7%)	15 (5.1%)	
I stayed in my home unable to work	37 (9.7%)	7 (8.1%)	30 (10.2%)	
I can work almost normally from my home	116 (30.5%)	29 (33.7%)	87 (29.6%)	
I can work for a few moments in my home	34 (8.9%)	6 (7.0%)	28 (9.5%)	
My job has been affected	32 (8.4%)	13 (15.1%)	19 (6.5%)	
Fired or my source income was suspended	25 (6.6%)	6 (7.0%)	19 (6.5%)	
My situation has not changed	62 (16.3%)	0 (0.0%)	62 (21.1%)	
I have been with prenatal/postnatal rest	6 (1.6%)	0 (0.0%)	6 (2.0%)	
MVPA (min/week)	583.0 ± 686.9	618.2 ± 629.9	572.7 ± 703.8	0.589
Physical fitness (score)	3.1 ± 0.8	3.4 ± 0.8	3.0 ± 0.8	**<0.001**
Sedentary time (min/day)	196.8 ± 155.4	220.9 ± 156.1	189.8 ± 154.7	0.102

MVPA: moderate-to-vigorous physical activity; bold values indicate statistical significance.

**Table 2 ijerph-19-01846-t002:** Baseline characteristics of active commuting, MVPA, physical fitness, and sedentary time by sex.

Parental Physical Characteristic	All (*n* = 380)	Fathers (*n* = 86)	Mothers (*n* = 294)	*p*-Value
Mode of commuting to work				
Stay home	68 (18.8%)	6 (7.4%)	62 (22.1%)	
Active	41 (11.4%)	12 (14.8%)	29 (10.4%)	**0.010**
Passive	252 (69.8%)	63 (77.8%)	189 (67.5%)	
Mode of commuting to home				
Stay home	73 (20.1%)	8 (9.9%)	65 (23.0%)	
Active	37 (10.2%)	11 (13.6%)	26 (9.2%)	**0.026**
Passive	253 (69.7%)	62 (76.5%)	191 (67.7%)	
MVPA (min/week)				
Public schools	601.9 ± 723.0	619.3 ± 7 45.8	597.0 ± 718.5	0.847
Private schools	553.7 ± 628.0	616.7 ± 417.4	534.3 ± 680.1	0.499
20–39 years	677.8 ± 792.3	763.4 ± 767.0	653.3 ± 800.6	0.458
40–59 years	509.4 ± 583.8	508.6 ± 482.7	509.7 ± 612.0	0.991
1 child	518.2 ± 544.6	734.8 ± 847.0	459.5 ± 413.3	**0.008**
≥2 children	632.2 ± 775.4	538.2 ± 413.3	661.2 ± 855.9	0.323
Physical fitness (score)				
Public schools	3.0 ± 0.8	3.4 ± 0.6	3.0 ± 0.8	**<0.001**
Private schools	3.3 ± 0.9	3.6 ± 0.9	3.2 ± 0.8	**0.017**
20–39 years	3.2 ± 0.8	3.6 ± 0.7	3.1 ± 0.7	**<0.001**
40–59 years	3.1 ± 0.8	3.3 ± 0.8	3.0 ± 0.8	**0.020**
1 child	3.1 ± 0.8	3.5 ± 0.6	3.0 ± 0.8	**<0.001**
≥2 children	3.1 ± 0.8	3.4 ± 0.8	3.1 ± 0.8	**0.015**
Sedentary time (min/day)				
Public schools	186.4 ± 149.4	197.2 ± 143.0	183.4 ± 151.4	0.560
Private schools	213.0 ± 163.5	255.5 ± 169.6	199.9 ± 160.0	0.078
20–39 years	184.7 ± 137.3	178.7 ± 109.4	186.4 ± 144.7	0.764
40–59 years	206.3 ± 167.7	252.8 ± 178.2	192.4 ± 162.5	**0.027**
1 child	205.5 ± 161.0	235.2 ± 149.2	197.5 ± 163.6	0.219
≥2 children	190.3 ± 151.0	211.1 ± 161.4	183.8 ± 147.6	0.260

MVPA: moderate-to-vigorous physical activity; bold values indicate statistical significance.

## Data Availability

The original database is available to researchers who require it. It can be requested directly at the following email address: fernando.rodriguez@pucv.cl.

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
