# Peer review of "Changes in Active Behaviours, Physical Activity, Sedentary Time, and Physical Fitness in Chilean Parents during the COVID-19 Pandemic: A Retrospective Study"

_ijerph, 2022, doi:10.3390/ijerph19031846_

Round 1

Reviewer 1 Report

Dear Authors,

In the attached document you can find the suggestions for improvement of this paper.

Best regards.

Author Response

Dear reviewer, thank you for the comments. 

Overall, we feel that the manuscript has improved very much. Below, you can find point-by-point comments and answers.

Thanks again.

Reviewer 2 Report

The study has some weaknesses but the most important is the limited content of the conclusions, which necessarily have to be deepened.
It is also suggested that the discussion be revised so that some data can be transferred to conclusions.

A new critical reading of the article is suggested, so that obvious statements or statements without scientific support are eliminated.

Author Response

(The authors gave the same response as above.)

Reviewer 3 Report

Thank you for the opportunity to review this manuscript. The authors have explored how the COVID-19 pandemic affected physical activity, sedentary behaviour and physical fitness in Chilean parents. The manuscript provides insights into how the pandemic influenced behaviours in a Chile. However, the data and discussions are weakened due to the writing style, which requires English proofing.   

Please see below for specific comments:

General

  1. The manuscript would benefit from English proof-reading as there are many cases where sentences are not clear and require amending

Abstract:

  1. Lines 18-19: Information is needed that data were collected to recall prior to the lockdown

Introduction

  1. General: The aim of the study specifies changes in active commuting, however very little focus is provided on this topic in the introduction. More details should be added to provide a background, such as the prevalence and modalities used in Chile
  2. Line 36: Please revise to ‘consequences have been devasting to…’
  3. Line 41: Please define COVID-19 since this is the first time it has been used as it was previously only referred to as SAR-CoV2 in the opening sentence
  4. Lines 44-45: A decrease in physical activity and increase in sedentary time can pose many health risks, and not just the physical fitness that is described. The authors should expand on this to ensure is it clear that this pattern of behaviour can affect many physical and mental outcomes, including physical fitness.
  5. Lines 71-72: The second aim is worded as if the authors already know the outcomes of the results, i.e., participants will have decreased their physical activity. At this stage the authors do not know this will be the case

Methods

  1. General: To gain a better context of the environment participants were living in during the period of data collection, information should be provided regarding the specific lockdown rules that were in place (e.g. specific movement restrictions) and how long these were in place for
  2. General: The methods section would be strengthened if the authors could provide a rationale for the choice of questionnaires used to obtain their data, such as validity metrics.
  3. Lines 82-83: How was this final sample size determined? Was an initial sample size calculation performed? Also, there is some disparity between male and female response numbers, did the authors consider this then when recruiting their study sample?
  4. Line 97-98: Please provide more specific information regarding the question participants were asked about how the pandemic changed their jobs. Is this referring to the location where they worked (e.g. forced to work at home), or the type of job they completed?
  5. Line 97: Did the authors collect information regarding the job roles participants were employed in. This should be added to provide a clear picture of the sample that were recruited.
  6. Line 99: Participants were asked to recall their activity levels ‘before’ the pandemic. Were participants provided with a specific time period to reflect upon? If some participant reflected on acutely prior to the pandemic (e.g., weeks), but other reflected over longer periods (e.g., months), this could cause disparity in the results.
  7. Line 99: Participant were asked to recall their activity levels ‘during’ the first lockdown. Were the lockdown rules consistent throughout the first lockdown period, or did they change based on factors such as Government announcements and case numbers? If changes occurred then this may have differentially influence participants activity levels and therefore their responses
  8. Line 114: The authors mention ‘the school type’ of children was used as a covariate. Please include how the school type data were obtained, as this has not been previously described in the previous section

Results

  1. Table 1: Are the MVPA and sedentary time data reported in this table the pre-lockdown data? This is not clear, please amend, for example including a more descriptive table title.
  2. Table 1: The use of ‘public school’ and ‘private school’ makes is seem as if this relates to the participants themselves. Instead it should be specified that this is their child’s school
  3. Lines 164-167: It is not clear where the data reported in the text compares to based on the values in the table. Please clarify or edit accordingly.
  4. Lines 177-178: Currently the reporting of the ‘differences’ is too vague. For example, it is not clear if the difference in ‘to work’ are referring to between the three condition (e.g. stay home, active or passive) for all participants, or based on the sex of the participants.
  5. Figure 1: Please alter the figures as often the bar charts exceed 100%, which should not be the case
  6. Figure 1: The figure reports data for during the lockdown, however the actual numerical values are not reported (as has occurred in Table 2 for the baseline/before lockdown data). For full clarity of data these should be added.
  7. Figure 2: The figure reports data for during the lockdown for MVPA, physical fitness and sedentary time, however the actual numerical values are not reported (as has occurred in Table 1 for the baseline/before lockdown data). For full clarity of data these should be added.

Discussion

  1. Line 246-248: A reference is needed for this sentence
  2. Line 248: Is this referring to older parents? Please clarify.
  3. Lines 264-265: This statement should be toned down since (i.e., ‘surely’) that authors have not measured the mode of commuting of their children
  4. Line 266: It is not clear what period is being referred to at the start of this sentence. Furthermore, it is not typical for a single sentence to form a paragraph. Instead this point should be embedded into the previous paragraph
  5. Lines 289-290: It is not typical for a single sentence to form a paragraph. Instead this point should be embedded into the previous paragraph
  6. Lines 308-309: It is not typical for a single sentence to form a paragraph. Instead this point should be embedded into the previous paragraph
  7. Limitations: The authors need to acknowledge limitations around the subjective assessment of activity behaviours and possible issues with recall error and response bias. The authors must also acknowledge limitations surrounding the self-reporting of fitness. The limitation that children’s activity data were not collected should also be included since the authors infer that the parents activity will influence their children’s but have no data to support this.

Conclusions

  1. Line 352: This sentence should not address the reader directly

Author Response

(The authors gave the same response as above.)

Round 2

Reviewer 3 Report

Thank you to the authors for taking the time to read the reviewer comments and amend the manuscript accordingly. The changes that have been made have significantly improved the manuscript, especially the readability of the work and presentation of the results.

I only have a couple of minor comments below:

  1. Line 99: Please report age to a whole number rather than 2 decimal places and this is not a meaningful value.
  2. Lines 111-113: it is not clear what is meant by capacity restrictions in closed spaces. Also, it is unclear what time period the phrase ‘currently’ is referring to. It could be assumed that currently is the present time.

Author Response

Dear reviewer. Thank you for your positive comments.

We have put a lot of energy into improving the paper and we liked it, especially thanks to your suggestions. We respond to your comments.

Line 99: Please report age to a whole number rather than 2 decimal places and this is not a meaningful value.

R: Thank you. the numbers have been replaced in both sexes.

Lines 111-113: it is not clear what is meant by capacity restrictions in closed spaces. Also, it is unclear what time period the phrase ‘currently’ is referring to. It could be assumed that currently is the present time.

R: Thank you for the comment. The restrictions at that time were regarding the capacity allowed in closed spaces, where few people were allowed for reasons of ventilation of those spaces. The sentence has been rewritten.
Regarding "currently" it has been delated and clarified to avoid confusion.

Thank you again for your comments.

Best regards,

The authors.